# Learned Meta-Tokens for Language Modeling

**Alok N. Shah**[1*]   **Khush Gupta**[1*]   **Keshav Ramji**[2*]   **Pratik Chaudhari**[1]

[1]University of Pennsylvania       [2]IBM Research AI
[*]Denotes equal contribution
{alokshah, khushg, pratikc}@upenn.edu, keshav.ramji@ibm.com

## Abstract

Transformer-based language models (LMs) notably struggle to reliably capture distant contextual information. This work introduces a novel approach using meta-tokens – special tokens injected during pre-training – paired with a dedicated meta-attention mechanism to guide LMs to use these tokens. We pre-train a language model equipped with meta-attention in addition to causal multi-head attention on <100B tokens, achieving strong performance on a suite of synthetic tasks. Our method facilitates length generalization up to $2\times$ the context window after extension with YaRN. We provide an information-theoretic analysis which reveals that meta-tokens *sharpen* the positional encoding, allowing them to operate as content-based anchors that compress preceding context and "cache" it within the meta-token. We empirically confirm this by visualizing model internals to study the residual stream. Together, our findings demonstrate that meta-tokens and meta-attention provide a simple, data-efficient pre-training method, grounded by mechanistic insights into their role in enabling length generalization behavior.

## 1 Introduction

Transformer-based language models (LMs) have showcased remarkable capabilities across diverse language tasks (Brown et al., 2020b; Chowdhery et al., 2022; OpenAI, 2023). Nevertheless, such models suffer from an inability to capture dependencies spanning over their entire context window. With growing adoption and ever-expanding demands on the context over which the model can process and reason, it is vital to develop methods that facilitate long-context adaptation and length generalization. Despite numerous architectural remedies, including sparse attention (Beltagy et al., 2020; Zaheer et al., 2020), recurrent blocks (Hutchins et al., 2022), and modified positional encoding (Press et al., 2021; Su et al., 2021; Chen et al., 2023), the fundamental challenge still remains: how can models reliably access and summarize distant context in a concise, cheap, yet expressive manner?

We propose a simple solution, by way of *meta-tokens*, learned tokens periodically injected into the input sequence during pretraining, and cleverly placed during fine-tuning. Unlike conventional dummy tokens (Goyal et al., 2024), meta-tokens are explicitly trained via a dedicated sparse attention layer, guiding the model to condense and "cache" contextual information as an in-line storage mechanism. As a result, these tokens act as adaptive content-based anchors, summarizing preceding context segments into compact representations. At inference time, meta-tokens provide implicit pathways to distant information, enabling models to generalize effectively across sequences longer than those encountered during training.

We demonstrate the empirical efficacy of this approach by pre-training a 152M parameter modified GPT-2 model with meta-tokens and a sparsely activated meta-attention mechanism. Our approach not only excels on recall-oriented synthetic tasks but also generalizes up to $2\times$ the context window (including after extension via YaRN) — a rare feat for decoder-only architectures trained on 100B tokens or less. We trace these gains to a subtle mechanism: meta-tokens provably induce a *sharpening* effect on positional encoding, enabling the meta-token to locate its position based on the content it stores and reducing the entropy of the attention distribution. We present theoretical and empirical evidence that this sharpening is responsible for the ability to retrieve information from relevant distant tokens, facilitating robust length generalization. Furthermore, by analyzing internal model activations and studying the rate-distortion tradeoff, we validate that meta-tokens function as compressed representations of context.

Our contributions can be summarized as follows:

1. We introduce a simple language model pre-training scheme using meta-tokens and a meta-attention mechanism to improve performance on a wide range of synthetic tasks.

2. We show that meta-tokens *sharpen* the positional encoding, enabling precise long-range attention; we further show that length generalization improves *without* positional encoding.

3. The *sharpening* hypothesis and implicit compression behavior are supported by visualizations of model internals and information-theoretic analysis into the rate-distortion tradeoff.

## 2 PRELIMINARIES

**Causal Multi-Head Attention.** Let $\mathbf{x} = \{x_1, x_2, \ldots, x_T\}$ denote an input sequence of tokens of length $T$, $\mathcal{V}$ denote the vocabulary size of V, and $E : \mathcal{V} \to \mathbb{R}^d$ represent the the token embedding function mapping each token to a $d$-dimensional vector. Each $x_t$ is embedded into some continuous representation where $\mathbf{e}_t = E(x_t) + \mathbf{p}_t$, such that $\mathbf{p}_t$ is the positional encoding for $t$.

In decoder-only architecture, we utilize causal self-attention to ensure that predictions for a given token are only based on preceding tokens. The causal self-attention mechanism modifies the attention computation by masking future positions in the attention weights. Formally:

$$\text{Causal Attention}(Q, K, V) = \text{softmax}\left(\frac{QK^\top}{\sqrt{d_k}} + M\right)$$

where $M$ masks future tokens, ensuring that the model can only attend to current and past tokens.

This masking zeros attention scores for future tokens, allowing only the relevant past tokens to influence the current token's representation.

**Positional Encoding.** Positional encoding was introduced in Transformer pre-training to provide models with information about the ordering of tokens. With absolute positional embeddings (APE; Vaswani et al. (2017)), each position $t$ in the sequence receives a vector $p_t$, independent of its content, so tokens are distinguished in an index-by-index manner. Given learned token-embedding lookup table $E : V \to \mathbb{R}^d$ for vocabulary $V$ and hidden dimension $d$, and positional embedding $p_t = \text{Emb}_{pos}(t)$ for $t \in [0, T - 1]$ and $\text{Emb}_{pos} \in \mathbb{R}^{T \times d}$. Each token embedding is then defined as $e_t = E(x_t) + p_t$; this method was used in GPT-2 and GPT-3 (Radford et al., 2019; Brown et al., 2020a).

By contrast, Rotary Position Embedding (RoPE; Su et al. (2023)) rotates each pair of embedding dimensions by an angle proportional to position, rather than adding a separate vector per position. This makes the difference in attention scores directly encode relative distance between embeddings. The hidden vector $h$ is split into $\frac{d}{2}$ contiguous 2-D slices, and the angle for a position $t$ is defined as $\theta_{t,i} = \frac{t}{10000^{2i/d}}$. The 2-D rotation matrix is taken as $R(\theta) = \begin{pmatrix} \cos\theta & -\sin\theta \\ \sin\theta & \cos\theta \end{pmatrix}$. Then, $\text{RoPE}(h)_t^{(2i:2i+1)} = R(\theta_{t,i})h^{(2i:2i+1)}$. This has proven successful in the Llama models (Grattafiori et al., 2024).

## 3 TRAINING LANGUAGE MODELS WITH META-ATTENTION

We introduce a set of $M$ meta-tokens (denoted as $m$); given a context length or block size of the model, $n$, we take $M = kn$ for some constant fraction $k \in [0, 1]$[1]. The aim of introducing these meta-tokens is to capture or store contextual information to enhance the model's retrieval and reasoning capabilities; attending to a meta-token should enable implicit retrieval of the context that it stores, guiding shortcut paths over the context window.

The $M$ tokens are injected into the input sequences during pre-training uniformly at random, which was informed by two key premises. While we desire interpretability and control in applying these tokens, and as a result, prefer distinguishability at the task level, this is challenging to do without

---

[1]We take $k = 0.1$ in practice; balancing next-token prediction over the standard vocabulary while injecting a non-trivial number of meta-tokens.

explicitly fixing a downstream task, impeding generality. The second consideration was in how they specifically they should be injected. While Zelikman et al. (2024) introduced <|startofthought|> and <|endofthought|> tokens interleaved between reasoning steps near punctuation (serving as natural break), the introduction of a rough periodicity between tokens during pre-training could result in being trapped into local minima in the optimization landscape. We instead chose to follow the random injection scheme, supported by the pre-training approach outlined in Goyal et al. (2024).

We ensure that the trained model incurs no loss for predicting meta-tokens, unlike a standard token in the vocabulary – the meta-tokens' indices are simply shifted and removed when computing the binary cross-entropy (BCE) loss.

**Meta-Attention Mechanism.** We augment our transformer $H$ to take $P$ which contains the positions of the meta-tokens. We introduce a sparse attention mechanism, called meta-attention, which selectively modifies attention scores for the specially marked "meta-tokens" within a sequence. This allows the model to simulate selective attention, influencing the final behavior by focusing on these meta-tokens. The underlying principles of the desired behavior is influenced by dual cross-attention (Jiang et al., 2024), such that operations are performed higher on the abstraction hierarchy than the feature space alone. This induces a meta-learning-like setup over which attention on the meta-tokens is learned.

Let the indices of meta-tokens be positions $\in \mathbb{N}^{B \times T'}$, where $T'$ is the number of meta-tokens per batch. We then build a meta mask $P \in \mathbb{R}^{B \times T \times T}$ to shape attention. For each batch $b$ and token pair $i, j$:

$$P[b, i, j] = \begin{cases} 0 & \text{if both } i \text{ and } j \text{ are meta tokens (i.e., } i, j \in \text{positions}[b, :]) \\ -\infty & \text{otherwise} \end{cases}$$

The meta-attention operation is defined as:

$$\text{MetaAttention}(Q, K, V) = \text{softmax}\left(\left(\frac{QK^\top}{\sqrt{d_k}} + M\right) + P\right)V$$

Where M is the same causal mask as before. Here, the meta mask $P$ allows attention to flow only among the meta tokens in the sequence, introducing a distinct interaction compared to regular attention. This meta-attention layer selectively modifies the attention by influencing the flow of information to and from these meta tokens, distinguishing itself from the standard causal attention.

To assemble the architecture used for our model, we insert the meta-attention mechanism after the causal masked self-attention computation, to specifically attend to the injected meta tokens, as defined above. We provide a complete breakdown of the architecture in Appendix A.

## 4 RESULTS

### 4.1 MODEL TRAINING AND ARCHITECTURE

All experiments were performed with 4 NVIDIA A100 GPUs, training the meta attention transformer on 98B tokens using Distributed Data Parallel (DDP) on the Colossal Cleaned Crawl Corpus (C4) (Raffel et al., 2020). The configuration and hyperparameters used in our pre-training are included in Appendix A and B. As a baseline, we also pre-train GPT-2 (124M) on C4, with identical hyperparameters. The primary change we make from a standard GPT-2 architecture is the addition of RoPE to enable better generalization to longer contexts and improve stability in next-token prediction tasks.

We extend our transformer model's context window from 1024 tokens to longer sequences by training two distinct models with context lengths of 4096 and 8192 tokens, respectively. This extension is implemented using the YaRN method (Peng et al., 2024), which dynamically scales Rotary Positional Embeddings (RoPE) to effectively process significantly longer sequences without compromising performance or computational efficiency. The key parameters are detailed in Appendix C.

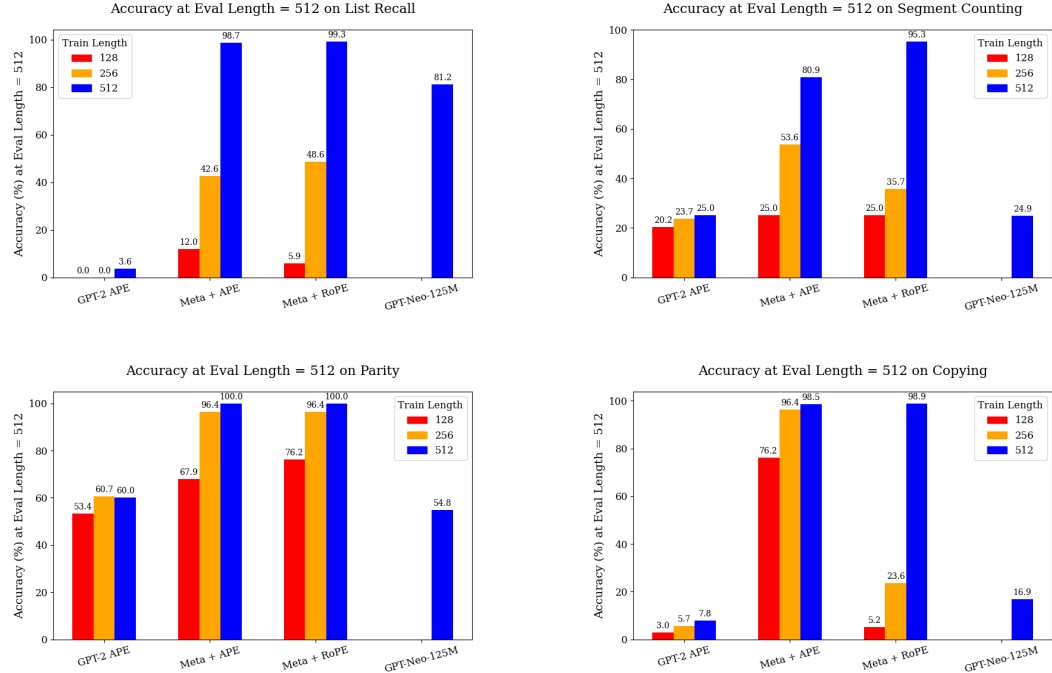

Figure 1: We study the performance of the pre-trained GPT-2 w/ APE, Meta-attention w/ APE, and Meta-attention w/ RoPE, as well as GPT-Neo-125M, all fine-tuned on synthetic data for their respective tasks at the maximum train lengths indicated in the legends. All experiments are performed on a test set of prompt lengths up to 512 tokens.

## 4.2 EXPERIMENTAL SETUP AND TASKS

We designed four synthetic tasks (*List Recall*, *Segment Counting*, *Parity*, and *Copying*) to probe the recall abilities of models trained with meta-tokens. Each task targets a distinct aspect of sequence memory and is offered in three difficulty levels determined by maximum sequence length. To guide meta-attention, we insert a designated _PAUSE_ meta-token at task-specific positions. We fine-tune models on synthetic data generated for these tasks (binned by instance length) and evaluate performance on a held-out test set. Examples of task instances are provided in Appendix H.

- **List Recall:** Given $N$ lists of length $k$, the model recalls a specified item. A _PAUSE_ token follows the target list and precedes the question. Difficulty varies with $N$ and $k$.

- **Segment Counting:** Lists contain a segment bounded by _PAUSE_ tokens. The prompt asks how often a target item appears within that segment. Difficulty scales with list number and size.

- **Parity:** A bit sequence contains a _PAUSE_ token marking a position. The model computes the XOR of preceding bits. Difficulty depends on sequence length.

- **Copying:** A text span is bracketed by _PAUSE_ tokens. The model must reproduce the span exactly. Difficulty grows with span length and complexity. Performance is measured by sequence accuracy.

Within these tasks, we study length generalization by fine-tuning in multiple phases. At each phase, we evaluate performance on sequences longer than those seen in training. Appendix D reports results at 2048 tokens, twice the pretraining length of 1024.

**Baselines.** For a controlled comparison, we also pre-train a GPT-2 model (NanoGPT, 124M; Karpathy (2023)) on C4, with identical hyperparameters as the meta-tokens model. Additionally, we use Eleuther AI's GPT-Neo-125M (Black et al., 2021) as another baseline.

Table 1: Token Accuracy (%) on List Recall and Segment Counting across long contexts.

| Task | Train/Finetune | 2k | 3k | 4k | 5k | 6k | 7k | 8k | 10k | 12k | 14k | 16k |
|------|----------------|----|----|----|----|----|----|----|-----|-----|-----|-----|
| List | 4k / 2k | 19.5 | 16.0 | 13.7 | 0.9 | 0.0 | 0.0 | 0.9 | 1.1 | 0.0 | 2.1 | 1.1 |
| | 4k / 4k | 85.0 | 88.2 | 90.2 | 20.5 | 1.8 | 1.0 | 3.5 | 4.4 | 1.1 | 2.1 | 2.1 |
| | 8k / 4k | 85.0 | 95.8 | 91.2 | 97.4 | 98.2 | 96.2 | 93.9 | 31.9 | 0.0 | 2.1 | 2.1 |
| | 8k / 8k | 92.9 | 98.3 | 97.1 | 100.0 | 98.2 | 100.0 | 100.0 | 89.0 | 26.1 | 10.4 | 9.6 |
| Count | 4k / 2k | 19.1 | 23.8 | 19.2 | 14.6 | 25.2 | 14.1 | 14.0 | 12.0 | 16.0 | 8.0 | 6.0 |
| | 4k / 4k | 17.5 | 23.8 | 31.8 | 20.3 | 30.4 | 19.3 | 19.1 | 14.0 | 26.0 | 12.0 | 16.0 |
| | 8k / 4k | 19.1 | 23.8 | 14.3 | 11.1 | 20.6 | 12.7 | 12.7 | 14.0 | 16.0 | 14.0 | 12.0 |
| | 8k / 8k | 27.0 | 33.3 | 15.9 | 19.1 | 27.0 | 19.1 | 23.8 | 22.0 | 18.0 | 18.0 | 18.0 |

Table 2: The configurations where zeroing the positional encoding at inference time results in accuracy improvements on the List Pointer task, denoted by the $\Delta$(pp) percentage points column.

| Model (Split, Train Len) | Full | No Pos | $\Delta$(pp) |
|--------------------------|------|--------|--------------|
| Meta + APE (medium, 128) | 77.8% | 88.9% | +11.1 |
| Meta + APE (hard, 128) | 11.1% | 22.2% | +11.1 |
| Meta + APE (extra-hard, 512) | 11.1% | 50.0% | +38.9 |
| Meta + RoPE (medium, 128) | 44.4% | 55.6% | +11.1 |
| Meta + RoPE (hard, 256) | 33.3% | 66.7% | +33.3 |
| Meta + RoPE (extra-hard, 256) | 0.0% | 22.2% | +22.2 |
| Meta + RoPE (extra-hard, 512) | 44.4% | 55.6% | +11.1 |

### 4.3 META-TOKENS IMPROVE PERFORMANCE ON SYNTHETIC RECALL-ORIENTED TASKS.

As seen in Figure 1, we find that the models trained on meta-tokens substantially outperform our pre-trained GPT-2 and GPT-Neo-125M baselines, across all tasks and all train lengths. The complete tables for these results are included in Appendix F.

Our models outperform the GPT-Neo-125M and GPT-2 with APE models by a substantial margin; given that GPT-Neo was pre-trained on 300B tokens, nearly three times the volume of data on which our meta-attention models were trained (albeit from a different corpus), highlighting the data-efficiency of our meta-tokens models. The models also gain in performance much more quickly with fine-tuning when increasing the train length – a phenomenon not observed with the GPT-2 models.

In addition to synthetic recall tasks, we evaluate our approach on the PG19 long-form language modeling benchmark. Training a meta-attention + RoPE model on 6B tokens from PG19 yields lower perplexity than both a matched GPT-2 baseline and an in-line retrieval long-context method, Landmark Attention (Mohtashami and Jaggi, 2023a) demonstrating that these gains transfer beyond synthetic settings. (Table 3)

To study the effect of positional encoding on our results, we ablate by zeroing out the positional encoding, zeroing out the text embedding, and performing both operations – all solely at the meta-token indices. Curiously, we observe in Tables 11-14 that the score *without* positional encoding nearly matches or exceeds the accuracy of the model *with* the positional encoding as is. The lone exception is the segment counting task, where there is a gap for all settings except the model trained with APE at a length of 256, which achieves a $+4.8\%$ improvement over the "Full" model. By contrast, zeroing out the token embedding hurts performance substantially in nearly every setting on List Recall, Segment Counting, and Copying; on Parity, this generally matches the performance of zeroing out the positional encoding. Thus, we find that 1. pre-training with meta-tokens and meta-attention boosts performance, and 2. zeroing out the positional encoding at just the meta-tokens can match or improve performance at inference time.

**Meta-Tokens Aid in Length Generalization.** In Figure 1 and Appendix F, we find that the model trained on meta-tokens length generalizes well on the parity and copying tasks with APE, and

Table 3: Perplexity on the PG19 dataset. Meta-Attention and GPT-2 are trained on 6B tokens from PG19 and Landmark Attention was trained on ~15B tokens.

| Model | PG19 PPL |
|---|---|
| GPT-2 (124M) | 16.13 |
| Landmark Attention | 16.23 |
| Meta-Attention + RoPE (Ours) | **14.79** |

performs somewhat well (much better than the baselines) on list recall and segment counting at a train length of 256. For instance, despite relatively similar performance at the 128 train length on the segment counting task, the performance on the test set of up to a length of 512 dramatically increases when training at the 256 length, by $+28.6\%$ with APE and $+10.7\%$ with RoPE, compared to $+3.5\%$ for GPT-2 with APE. Table 1 exhibits a similar trend for the YaRN models, achieving strong performance across its respective context windows, and even achieves non-trivial accuracy beyond the window. Fine-tuning the 8k YaRN model on examples of up to a length of 4k can generalize very well up to 8k. These findings underscore the substantial advantages of training with meta-tokens and the nuanced role positional encoding plays in task-specific and length-generalization contexts.

Moreover, when looking at the results on Meta + RoPE on test set lengths of prompts up to 1024 tokens (denoted extra-hard in Table 2), we find that zeroing out the positional encoding also plays a sizable role in improving length generalization, especially in the List Recall task. While the model originally achieves performances of $11.1\%$, $0\%$ and $44.4\%$ when fine-tuned on train lengths of 512 (APE), 256 and 512 (RoPE), respectively, the scores improve by $+38.9\%$, $+22.2\%$ and $+11.2\%$, by simply zeroing out the positional encoding at the meta-tokens.

If a meta-token retains its PE, a portion of its representational capacity is spent encoding position rather than semantic content. This index-dependent signal introduces unnecessary variance, increasing the distortion of the compressed summary. By contrast, zeroing out the PE forces the full embedding capacity to encode task-relevant information. As a result, we observe lower distortion (higher retrieval accuracy) at a given rate—both theoretically and empirically—across all four synthetic tasks.

## 5 WHAT MAKES META-TOKENS USEFUL?

The results in Table 2 suggest that the positional encoding of the meta-token can potentially be holding back the downstream performance of the meta-attention models. We posit that the model is instead relying on its content – cached context stored within the meta-token – to *sharpen* its sense of its position in the sequence.

Next, we aim to formally define this notion of sharpness in the context of positional encoding, and its relationship to the model's logits. Let $\alpha_{i\rightarrow k} = \text{softmax}_k(Q_i K_j^T + b_{i-j})$ be the attention distribution for query $i$ over keys $j$, with relative bias term $b_{i-j}$. We define the *sharpness* of the positional encoding by the entropy:

$$H(\alpha_i) = -\sum_j \alpha_{i\rightarrow j} \log \alpha_{i\rightarrow j}$$

Intuitively, when a meta-token is present at position $t$, the model's attention becomes peaked around a small set of keys; this "honing in" behavior reduces $H(\alpha)$ compared to APE or RoPE without meta-tokens. In this manner, meta-tokens behave as **content-driven landmarks** – they serve as a low-entropy channel that serves as a pointer to relevant context. As noted prior, the data efficiency observation suggests that the meta-token helps to accelerate next-token prediction behavior while introducing a stabilizing effect in the midst of noisy positional encoding.

**Theorem 5.1.** *Consider a Transformer head at query position $i$ over keys $1, \ldots, N$. Let $\alpha_i^{abs}(j) \propto \exp(Q_i K_j^T)$ be the attention under absolute positional encoding and let $\alpha_i^{meta} \propto \exp(Q_i K_j^T + \delta_{j,j^*}\Delta)$ when a meta-token at position $j^*$ introduces an additive logit boost of $\Delta > 0$. Then, for some function $\kappa(\Delta) > 0$:*

$$H(\alpha_i^{meta}) \leq H(\alpha_i^{abs}) - \kappa(\Delta) \tag{1}$$

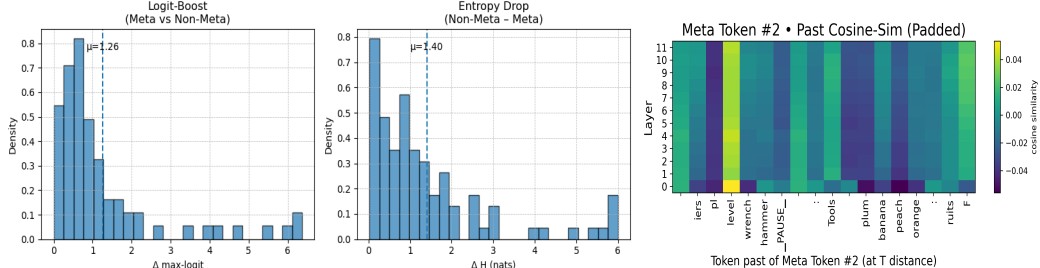

Figure 2: (Left) We analyze the change in logits at the meta-token position, and observe that the meta-tokens do indeed induce a sizable boost in logits compare to zeroing its token embedding. (Middle) We find the boost in logits to correspond with a meaningful reduction in Shannon entropy over the softmax of the logits between the zeroed meta-token sequence and the sequence with the meta-token as is. This corroborates with our assumptions and claims in Theorem 5.1. (Right) We study the implicit "caching" ability of the meta-token by studying the cosine similarity over the token embeddings. We observe high spikes (the yellow column), diminishing as we move further away. This substantiates our claims of the presence of an implicit compression and "caching" mechanism.

*Proof Sketch.* Parametrize the path by $t \in [0, \Delta]$, and define logits $\ell_j^{(t)}$ and their softmax $\alpha^{(t)}$ respectively. Since boosting the true index tightens the margin, the derivative of $H(\alpha^{(t)}$ is strictly negative. Therefore, over the path, $H(\alpha^{(\Delta)}) < H(\alpha^{(0)})$, so $H(\alpha^{\text{meta}}) < H(\alpha^{\text{abs}})$, where their difference must be a function of $\Delta$. The full proof is included in Appendix G. □

We note that this theorem also applies to RoPE, using $\alpha_i^{\text{RoPE}}(j) \propto \exp Q_i (\text{RoPE}(K_j))^T$. A natural consequence of Theorem 5.1 is that the meta-token operates as an "anchor" from a logits standpoint by creating a margin $\Delta$ that concentrates the softmax. Concretely, we can specify that for meta-token $m_t$ at position $t$ and embedding $e_t \in \mathbb{R}^d$, and query at position $i$ with vector $Q_i$, has contribution to the $(i, j)$ logit of $\Delta_{i,j}^{(t)} = Q_i \cdot W e_t \times \mathbf{1}_{j=t}$ for learned linear head $W$. Summing over $t$ yields the bias matrix $B \in \mathcal{B}_{\text{meta}}$, the set of all realizable bias matrices under the meta-token embeddings. Thus, any learned meta-token embedding – provided that it adds to the logits at the summary position $j^*$ – guarantees sharper attention by reducing that attention head's entropy.

In Figure 2, we analyze the logits, comparing two settings: (1.) the current meta-token and (2.) the meta-token with its token embedding zeroed out. We find that the former gains a sizable amount over the latter, reinforcing the assumption made in Theorem 5.1 that the meta-token introduces an additive logit boost of $\Delta > 0$. Our empirical results show that the entropy over the softmax distribution of the logits decreases (the difference between "non-meta-token" and "meta-token" is positive), thus corroborating our central claim in Theorem 5.1.

## 5.1 A RATE-DISTORTION PERSPECTIVE ON CONTEXT COMPRESSION

Given that these results provide evidence that meta-tokens can compress context in their representation, we develop mathematical formalizations to analyze this behavior. In particular, we turn to information-theoretic tools – specifically, an information bottleneck view.

For a meta-token at $x_m$ succeeding a sequence of tokens $X = x_{i:m-1}$ from indices $i$ to $m - 1$, we consider a compression function $\zeta(\cdot)$ which transforms the subsequence $X$ into $x_m$. As such, we define $\hat{X} = \zeta(X) = \zeta(x_{i:m-1})$ to be the *compressed representation stored* in $x_m$. This can be generalized to the full set of $M$ meta-tokens:

$$\hat{X}_{1:M} = [\zeta_1(X_{1:m_1-1}), \zeta_2(X_{m_1+1:m_2-1}), \ldots \zeta_M(m_{M+1} : m_n)]$$

For practicality, we consider the variational information bottleneck (Alemi et al., 2017). This introduces an encoder $q_\phi(\hat{x} \mid x)$ and decoder $q_\theta(y \mid \hat{x})$, along with a simple prior $r(\hat{x})$ (e.g. $N(0, 1)$), yielding the following form to solve for these variational distributions:

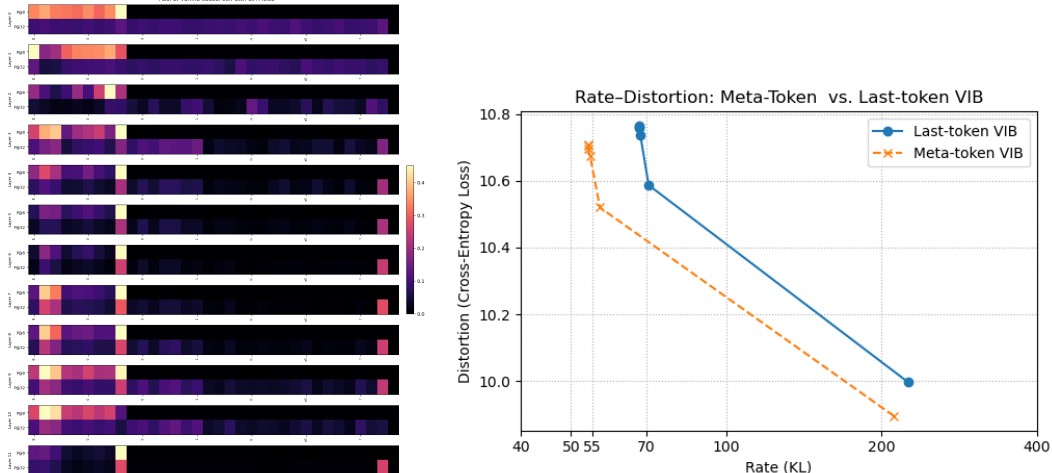

Figure 3: (Left) This plot shows the residual stream at each layer, highlighting the meta-token's role in causal attention. The colors before the meta-token (colored band across layers) indicate the context it attends to/stores. The rightmost line shows the final meta-token, which attends to the previous one at the band. (Right) We analyze the variational information bottleneck (VIB) objective and its decomposition into rate and distortion components. Supporting Theorem 5.1, for a given rate $R$, the distortion $D$ is strictly lower for the meta-token compared to the last non-meta-token in the sequence.

$$\min_{q_\phi, q_\theta} \mathbb{E}_{p(x,y)} \big[ \mathbb{E}_{q_\phi(\hat{x}|x)} [-\log q_\theta(y \mid \hat{x})] + \beta \cdot \mathbb{E}_{p(x)} [KL(q_\phi(\hat{x} \mid x) \| r(\hat{x}))]$$

This form admits an equivalent perspective in rate-distortion theory. Specifically, the first term measures the quality in predicting the downstream target given a lossy compression $\hat{X}$ ("distortion"). The second term measures the average number of bits required to encode $\hat{X}$, relative to some simple reference code $r(z)$ ("rate"). As such, analyzing rate-distortion curves – sweeping over values of $\beta$ – can provide valuable insights into the quality of the "compression" behavior and its informativeness when the meta-token is attended to.

**Theorem 5.2.** *Let $D_{abs}(R)$ be the minimum distortion achievable at rate $R$ under the VIB objective only using absolute positional encoding (no meta-tokens), and let $D_{meta}(R)$ be the minimum distortion achievable at rate $R$ with meta-tokens. Then, for every $R \geq 0$,*

$$D_{meta}(R) \leq D_{abs}(R) \tag{2}$$

Intuitively, meta-tokens expand the feasible set of encoders and decoders, which will either match or lower distortion for a given rate. Thus, the quality of compression with respect to its informativeness in predicting the target can only improve. In Appendix G.4, we provide further intuition on how meta-tokens facilitate context caching by showing that they act as *inducing points* in a Nyström approximation of the softmax kernel. This connects directly to our rate-distortion analysis, as we bound the prediction error (distortion) by the quality of kernel approximation.

## 5.2 RATE-DISTORTION INFORMS THE QUALITY OF CONTEXT CACHING

To obtain empirical rate–distortion curves for our meta-token bottleneck in Figure 3, we freeze the pre-trained meta-token model and fix a small variational bottleneck head to the *last* meta-token hidden state. Concretely, let $h_m \in \mathbb{R}^D$ be the output of the final Transformer layer at the last meta-token position. We introduce

$$q_\phi(z \mid h_m) = \mathcal{N}\big(\mu_\phi(h_m), \, \text{diag}(\sigma_\phi^2(h_m))\big), \quad q_\theta(y \mid z) = \text{softmax}(Wz + b),$$

with $\mu_\phi, \sigma_\phi : \mathbb{R}^D \to \mathbb{R}^L$ and $W \in \mathbb{R}^{|\mathcal{V}| \times L}$. We then optimize the ELBO:

$$\min_{\phi, \theta} \mathbb{E}_{h_m, y} \big[ -\log q_\theta(y \mid z) \big] + \beta \, \mathbb{E}_{h_m} \big[ \text{KL}\big(q_\phi(z \mid h_m) \, \| \, \mathcal{N}(0, I)\big) \big].$$

Training is performed on the **small** List-Pointer D.1.1 split (50 examples, batch size 1), for 5 epochs at each $\beta \in \{0.01, 0.02, 0.05, 0.1, 0.2, 0.5, 1.0\}$. After each run, we record the *average* cross-entropy loss ("distortion") and KL ("rate") on the same 50 examples. Finally, we plot the resulting rate–distortion curves on a `symlog` x-axis (linear below 20 nats, logarithmic above) so that both the low-rate "knee" and the long tail are visible (see Figure 3).

## 6  RELATED WORK

**Pause and Memory Tokens**    As detailed in our work, recent studies on Transformer-based models have explored the introduction of special tokens, beyond ordinary vocabulary symbols. *Pause* or *dummy* tokens as introduced in Goyal et al. (2024) enhance computational width, allowing models to perform additional internal computation by effectively delaying their outputs, yielding empirical gains on question answering and reasoning-intensive tasks. Similarly, Pfau et al. (2024) explore using filler tokens – sequences of seemingly meaningless symbols – as a stand-in for chain-of-thought. Recent works such as Merrill and Sabharwal (2025) and London and Kanade (2025) demonstrate that the inclusion of a polynomial number of pause tokens to constant-depth Transformers increases expressivity to represent the $TC^0$ class. In contrast, our work introduces meta-tokens that are not merely delays or placeholders but are explicitly designed to direct attention and structure computation, allowing the model to recall and generalize over the context length in ways these prior approaches do not address. Works such as Memory Transformer (Burtsev et al., 2021) and Landmark Attention (Mohtashami and Jaggi, 2023b) introduce memory tokens; the former prepends them, while the latter uses them as learnable keys for retrieval over blocks of context. Unlike Memory Transformer's prepended slots or Landmark Attention's block-level landmarks, our meta-tokens are dynamically positioned and operate through a dedicated meta-attention layer rather than fixed structural scaffolds.

**Positional Encoding**    We have already described absolute positional embeddings (APE), rotary positional embeddings (RoPE) and relative bias in Section 2. In addition to these methods, ALiBi (Press et al., 2022) adds a fixed linear penalty to attention scores based on the distance between query and key positions, favoring nearer tokens and generalizing to longer contexts with minimal loss in perplexity. Recent work has suggested that Transformers without any added position embeddings can still learn order information and, in some cases, generalize to longer sequences better than models with standard positional encoding. NoPE (Kazemnejad et al., 2023) showed that models trained without positional embeddings can achieve strong length extrapolation in comparison to models trained with positional encoding. They can internally represent both absolute and relative PEs without any explicit positional signal, suggesting these may emerge implicitly via training dynamics or over the data distribution. NoPos (Haviv et al., 2022) also found a similar result, suggesting that models trained without PE can infer their absolute position due to causal attention masks. These findings are highly relevant to our work, given our evidence on length generalization behavior whiling zeroing the positional encoding at the meta-tokens.

## 7  DISCUSSION AND LIMITATIONS

Our findings suggest that decoder-only language models trained with meta-tokens and meta-attention achieve strong performance on recall-oriented tasks. Furthermore, they are able to length generalize, with performance improvements when removing the effect of positional encoding at the meta-tokens. Given the prior findings of NoPos, we believe the introduction of the meta-attention mechanism and a second causal mask (the "meta mask") could be responsible for this behavior, provided that this behavior is specific to the meta-tokens. We suggest that hybrid attention methods such as RNoPE (Yang et al., 2025) could be suitable for facilitating long-context modeling with meta-tokens.

Given the findings that the meta-tokens operate like anchors within the context, it would be valuable to explore the impact of our proposed mechanism in pre-training larger models over longer context windows, under greater computational resources. We employ synthetic tasks that are well-aligned to recall abilities, and design experiments to test length generalization, with the aim of strong synergy with long-context modeling capabilities. Nonetheless, training larger models would indicate the viability of our approach for real-world deployment.

Notably, our method requires little overhead – the addition of meta-tokens is a simple data augmentation strategy, and the meta-attention layer is added after standard causal masked self-attention, as described in Appendix A. It would also be informative to study larger-scale corpora – given the data-efficient nature of the meta-tokens approach in vastly outperforming the vanilla GPT-2 model at the $\approx 100B$ tokens scale, how rapidly does each model saturate our designed synthetic tasks?

## 8    CONCLUSION

We introduce *meta-tokens* in language model pre-training, in addition to a dedicated meta-attention mechanism which learns the relationship between standard tokens and meta-tokens. We find that this improves performance on a suite of synthetic recall tasks, and enables length generalization behavior when removing the positional encoding at each meta-token. We provide evidence to suggest that the meta-tokens sharpen the positional encoding, enabling them to operate as content-based landmarks in the context; we further show that they implicitly compress preceding context, demonstrated by similar token embeddings. These interesting phenomena demonstrate the promise of long-context language modeling enabled via data-efficient pre-training using meta-tokens.

## ACKNOWLEDGEMENTS

We thank Surbhi Goel for valuable discussions that shaped the direction of this study and for support with computational resources. We also thank Ben Keigwin for helpful suggestions in designing our analysis. We are grateful to Simran Arora for feedback that led to substantial revisions of earlier drafts. This work was supported in part by a research compute grant from Lambda Labs.

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

# A  FULL ARCHITECTURE DETAILS

We provide a full outline of the architecture design out method uses. Our architecture is equivalent to the NanoGPT (GPT-2) architecture, while introducing the meta-attention block after the initial causal masked attention and layer normalization computation.

1. **Input Layer:** Given an input sequence of tokens $\mathbf{x} = \{x_1, x_2, \ldots, x_T\}$, we first embed each token into a continuous representation. Instead of absolute positional encodings, we apply Rotary Position Embeddings (RoPE) Su et al. (2023) to inject positional information. For each token, the embedded representation is:

$$\mathbf{e}_t = \text{RoPE}(E(x_t), t),$$

   where $\text{RoPE}(\cdot, t)$ denotes the rotary positional embedding applied to the $t^{\text{th}}$ position, with a base $\theta = 10000.0$.

2. **Causal Masked Self-Attention:** The first layer consists of the causal masked self-attention mechanism. For each head $h$, the attention operation is computed as:

$$\text{CausalAttention}_h(Q, K, V) = \text{softmax}\left(\frac{QK_h^\top}{\sqrt{d_k}} + M\right) V_h,$$

   where $Q, K, V$ are the query, key, and value matrices derived from the input embeddings $\mathbf{E}$, and $M$ is the mask matrix.

3. **Meta Attention Layer:** After the causal masked self-attention, we integrate the meta-attention mechanism to specifically attend to the injected meta tokens. This operation is defined as:

$$\text{MetaAttention}(Q, K, V, P) = \text{softmax}\left(\frac{QK^\top}{\sqrt{d_k}} + M_{\text{causal}} + P\right) V,$$

   where $P$ is the meta mask constructed from the indices of the meta tokens.

4. **Feedforward Layer:** Following the attention layers, we pass the output through a feedforward neural network defined by:

$$\text{FFN}(x) = \text{ReLU}(xW_1 + b_1)W_2 + b_2,$$

   where $W_1, W_2$ are weight matrices, and $b_1, b_2$ are bias vectors.

5. **Layer Normalization:** After both the causal self-attention and meta-attention operations, we apply layer normalization:

$$\text{LayerNorm}(x) = \frac{x - \mu}{\sigma + \epsilon},$$

   where $\mu$ and $\sigma$ are the mean and standard deviation of the features, and $\epsilon$ is a small constant for numerical stability.

6. **Final Output Layer:** The final layer projects the output of the last feedforward layer back to the vocabulary size to produce the s for the next token prediction:

$$\text{s} = \text{softmax}(xW_{\text{out}} + b_{\text{out}}),$$

   where $W_{\text{out}}$ and $b_{\text{out}}$ are the output weight matrix and bias vector, respectively.

## B PRE-TRAINING HYPERPARAMETERS AND MODEL DETAILS

Our decoder-only modified GPT-2 model was pre-trained on the C4 dataset with the following configuration and hyperparameters:

Table 4: Pretraining Configuration Parameters

| Parameter | Value |
|---|---|
| Batch Size | 12 |
| Gradient Accumulation Steps | 40 |
| Block Size | 1024 |
| Number of Layers | 12 |
| Number of Heads | 12 |
| Embedding Size | 768 |
| Learning Rate | 6e-4 |
| Weight Decay | 1e-1 |
| Max Iterations | 600,000 |
| Warmup Iterations | 2,000 |
| Minimum Learning Rate | 6e-5 |
| Dropout Rate | 0.0 |
| RoPE Theta | 10000.0 |
| Initial Model | Resume |
| Optimizer | AdamW |
| AdamW Beta1 | 0.90 |
| AdamW Beta2 | 0.95 |
| Gradient Clipping | 1.0 |
| Tokenizer | tiktoken |

## C YARN HYPERPARAMETERS

| Parameter | 4096-token model | 8192-token model |
|---|---|---|
| yarn_scale | 4.0 | 8.0 |
| yarn_original_max_seq_len | 1024 | |
| yarn_extrapolation_factor | 1.0 | |
| yarn_attn_factor | 1.0 | |
| yarn_beta_fast | 32.0 | |
| yarn_beta_slow | 1.0 | |

Table 5: YaRN parameter configurations for extended context models.

## D ADDITIONAL EXPERIMENTAL DETAILS

### D.1 SYNTHETIC DATA GENERATION

We generate 90,000 train examples and held-out test set of 10,000 examples for each task.

#### D.1.1 LIST RECALL

We generate a suite of "list-pointer" examples by sampling random categories and list items, inserting a special meta token as a marker, and asking the model to recover the item immediately following the meta-token. Each example consists of:

1. m categories drawn without replacement from a fixed set of 20.

2. n items per category, sampled with replacement from the category's 10–item inventory

3. One "target" category in which we inject a single meta token after the jth item ($j \in [n]$) and then append the remaining items

4. A question line "Q: What is item j of <target>? _META_"

This pipeline yields curriculum-structured data that systematically probes the model's ability to attend to and copy items in long, multi-list contexts.

Table 6: Curriculum schedule for synthetic data.

| Phase | m (Num. categories) | n (List length) | Approx. prompt-token range |
|:-----:|:-----:|:-----:|:-----:|
| 1 | Uniform 3–8 | Uniform 3–10 | Short ($\approx$ 100–200 tokens) |
| 2 | Uniform 8–12 | Uniform 3–16 (bimodal) | Mid-range ($\approx$ 200–300 tokens) |
| 3 | Uniform 12–19 | Mixture $\{3\text{–}8, 9\text{–}16, 17\text{–}25\}$ | Full-range ($\approx$ 500–700 tokens) |
| 4 | Uniform 15–20 | Uniform 40–60 | "Extra-hard" $\leq 1024$ tokens |
| 5 | Uniform 15–20 | Uniform 90–110 | "Long" $\leq 2048$ tokens |

### D.1.2 SEGMENT COUNTING

Similar to List-pointer, except the model must count occurrences of a target token within a meta-token bracketed list segment. Uses the schedule dictated by Table 6. Asks the question: "Q: How many times does <token> appear between the pauses around <Category>? _META_".

### D.1.3 PARITY

Generates examples where the model computes the XOR (parity) of a bit-string segment up to the first L characters where L is drawn phase-dependently. the same scheduling dictated by Table 6. Asks the question: "Q: What is the XOR of all bits before this pause? _META_ "

### D.1.4 COPYING

Generates examples where the model must copy a bracketed span from a text. Uses schedule dictated by Table 6 and samples an additional copy length $C$ and distance length $D$ depending on the phase

### D.1.5 MULTI-HOP

Generates examples where the model must resolve a chain of references across list indices. Each example begins with an input list containing items at numbered positions. The task specifies a starting index, and the model must repeatedly follow the indicated pointers. Uses schedule dictated by Table 6.

## E LICENSES

### NANOGPT

Our implementation of the vanilla GPT-2 is based on the nanoGPT repository (https://github.com/karpathy/nanoGPT), which is licensed under the MIT License.

### ELEUTHERAI GPT-NEO-125M

We directly use the EleutherAI GPT-Neo 125M model checkpoint and weights, available via the Hugging Face Model Hub at https://huggingface.co/EleutherAI/gpt-neo-125m. This model is released under the MIT License.

### C4 DATASET

Our model was trained on the C4 dataset (https://huggingface.co/datasets/allenai/c4), which is provided under the Open Data Commons Attribution License (ODC-BY).

TIKTOKEN

We use the `tiktoken` library from OpenAI for tokenization ([https://github.com/openai/tiktoken](https://github.com/openai/tiktoken)), which is released under the MIT License.

# F  Complete Experimental Results

## F.1  Synthetic Task Accuracies Across Test Lengths

We stress test RoPE models at a sequence length of 2048—twice the pretraining block size of 1024—as relative position embeddings naturally support extrapolation beyond the training context window. In contrast, absolute positional encodings (APE) cannot generalize to sequences longer than those seen during pretraining.

Table 7: Accuracy (%) across evaluation lengths for each model train on List Recall

| Model (Train Length) | 128 | 256 | 512 | 1024 | 2048 |
|---|---|---|---|---|---|
| GPT-2 APE (128) | 4.2 | 1.2 | 0.0 | 0.0 | — |
| GPT-2 APE (256) | 6.8 | 2.4 | 0.0 | 0.0 | — |
| GPT-2 APE (512) | 19.8 | 9.5 | 3.6 | 0.0 | — |
| Meta + APE (128) | 100.0 | 86.4 | 12.0 | 4.1 | — |
| Meta + APE (256) | 100.0 | 98.6 | 42.6 | 3.9 | — |
| Meta + APE (512) | 100.0 | 100.0 | 98.7 | 11.1 | — |
| Meta + RoPE (128) | 100.0 | 60.7 | 5.9 | 0.0 | 0.0 |
| Meta + RoPE (256) | 100.0 | 100.0 | 48.6 | 23.5 | 0.0 |
| Meta + RoPE (512) | 100.0 | 100.0 | 99.3 | 58.9 | 5.6 |
| GPT-Neo-125M | 85.6 | 86.0 | 81.2 | — | — |

Table 8: Accuracy (%) across evaluation lengths for each model trained on Segment Counting. Each model is evaluated on longer contexts than seen during training.

| Model (Train Length) | 128 | 256 | 512 | 1024 | 2048 |
|---|---|---|---|---|---|
| GPT-2 APE (128) | 32.1 | 27.4 | 20.2 | 0.0 | — |
| GPT-2 APE (256) | 40.3 | 56.2 | 23.7 | 0.0 | — |
| GPT-2 APE (512) | 30.1 | 32.1 | 25.0 | 0.0 | — |
| Meta + APE (128) | 77.4 | 55.9 | 25.0 | 11.1 | — |
| Meta + APE (256) | 83.3 | 77.4 | 53.6 | 22.4 | — |
| Meta + APE (512) | 91.7 | 79.8 | 80.9 | 33.3 | — |
| Meta + RoPE (128) | 77.4 | 64.3 | 25.0 | 22.7 | 0.0 |
| Meta + RoPE (256) | 64.3 | 64.3 | 35.7 | 33.3 | 0.0 |
| Meta + RoPE (512) | 90.9 | 91.4 | 95.3 | 66.7 | 11.1 |
| GPT-Neo-125M | 31.4 | 25.9 | 24.9 | — | — |

Table 9: Accuracy (%) across evaluation lengths for each model train on Parity

| Model (Train Length) | 128 | 256 | 512 | 1024 | 2048 |
|---|---|---|---|---|---|
| GPT-2 APE (128) | 75.0 | 56.0 | 53.4 | 45.2 | — |
| GPT-2 APE (256) | 75.0 | 67.0 | 60.7 | 46.2 | — |
| GPT-2 APE (512) | 75.0 | 54.8 | 60.0 | 40.5 | — |
| Meta + APE (128) | 100.0 | 75.0 | 67.9 | 52.4 | — |
| Meta + APE (256) | 100.0 | 97.6 | 96.4 | 69.1 | — |
| Meta + APE (512) | 100.0 | 100.0 | 100.0 | 86.7 | — |
| Meta + RoPE (128) | 100.0 | 66.7 | 76.2 | 59.5 | 44.1 |
| Meta + RoPE (256) | 97.6 | 100.0 | 96.4 | 61.9 | 52.4 |
| Meta + RoPE (512) | 100.0 | 100.0 | 100.0 | 69.1 | 63.1 |
| GPT-Neo-125M | 80.4 | 59.1 | 54.8 | — | — |

Table 10: Accuracy (%) across evaluation lengths for each model trained on Copying

| Model (Train Length) | 128 | 256 | 512 | 1024 | 2048 |
|---|---|---|---|---|---|
| GPT-2 APE (128) | 6.0 | 5.3 | 3.0 | 0.0 | — |
| GPT-2 APE (256) | 6.8 | 6.0 | 5.7 | 0.0 | — |
| GPT-2 APE (512) | 3.8 | 4.8 | 7.8 | 0.0 | — |
| Meta + APE (128) | 100.0 | 66.7 | 76.2 | 2.6 | — |
| Meta + APE (256) | 100.0 | 100.0 | 96.4 | 7.9 | — |
| Meta + APE (512) | 100.0 | 100.0 | 98.5 | 87.4 | — |
| Meta + RoPE (128) | 96.6 | 73.0 | 5.2 | 0.0 | 0.0 |
| Meta + RoPE (256) | 98.2 | 100.0 | 23.6 | 9.3 | 3.2 |
| Meta + RoPE (512) | 99.0 | 98.9 | 98.9 | 89.4 | 11.8 |
| GPT-Neo-125M | 31.5 | 22.7 | 16.9 | — | — |

## F.2 ABLATIONS ON POSITIONAL ENCODING AND TOKEN EMBEDDING

Table 11: Accuracy (%) on the List-Recall task under different ablations: zeroing the positional encoding (No Pos), zeroing the text embeddings (No Embed), or zeroing both of the meta-tokens.

| Model (PE) | Full | No Pos | No Embed | Neither |
|---|---|---|---|---|
| Meta + APE (128) | 100.0 | 99.3 | 17.4 | 59.7 |
| Meta + RoPE (128) | 100.0 | 100.0 | 32.4 | 24.0 |
| Meta + APE (256) | 86.4 | 86.9 | 12.2 | 16.2 |
| Meta + RoPE (256) | 100.0 | 100.0 | 4.0 | 6.6 |
| Meta + APE (512) | 100.0 | 100.0 | 52.1 | 84.3 |
| Meta + RoPE (512) | 100.0 | 100.0 | 59.6 | 25.2 |

Table 12: Accuracy (%) on the Segment Counting task under different ablations: zeroing the positional encoding (No Pos), text embeddings (No Embed), or both, only on the meta-token.

| Model (Train Length) | Full | No Pos | No Embed | Neither |
|---|---|---|---|---|
| Meta + APE (128) | 77.4 | 63.1 | 31.0 | 47.6 |
| Meta + APE (256) | 83.3 | 88.1 | 32.1 | 40.5 |
| Meta + APE (512) | 91.7 | 82.1 | 34.5 | 51.2 |
| Meta + RoPE (128) | 77.4 | 70.2 | 59.5 | 36.9 |
| Meta + RoPE (256) | 64.3 | 53.6 | 30.9 | 30.9 |
| Meta + RoPE (512) | 80.9 | 72.6 | 36.9 | 25.0 |

Table 13: Accuracy (%) on the Parity task under different ablations: zeroing the positional encoding (No Pos), text embeddings (No Embed), or both, only on the meta-token.

| Model (Train Length) | Full | No Pos | No Embed | Neither |
|---|---|---|---|---|
| Meta + APE (128) | 100.0 | 100.0 | 100.0 | 100.0 |
| Meta + APE (256) | 75.0 | 77.4 | 77.4 | 79.8 |
| Meta + APE (512) | 67.9 | 71.4 | 72.6 | 66.7 |
| Meta + RoPE (128) | 100.0 | 97.6 | 100.0 | 100.0 |
| Meta + RoPE (256) | 66.7 | 66.7 | 73.8 | 66.7 |
| Meta + RoPE (512) | 76.2 | 75.0 | 75.0 | 64.3 |

Table 14: Accuracy (%) on the Copying task under different ablations: zeroing the positional encoding (No Pos), text embeddings (No Embed), or both, only on the meta-token.

| Model (Train Length) | Full | No Pos | No Embed | Neither |
|---|---|---|---|---|
| Meta + APE (128) | 96.6 | 93.2 | 7.2 | 4.8 |
| Meta + APE (256) | 98.2 | 99.6 | 5.0 | 3.6 |
| Meta + APE (512) | 99.0 | 96.6 | 5.7 | 5.4 |
| Meta + RoPE (128) | 100.0 | 99.6 | 6.9 | 4.9 |
| Meta + RoPE (256) | 100.0 | 100.0 | 4.5 | 5.1 |
| Meta + RoPE (512) | 100.0 | 95.6 | 6.9 | 4.9 |

### F.3 POSITIONAL ENCODING ROBUSTNESS ABLATIONS

Table 15: Accuracy (%) on the List Pointer task with Gaussian noise added to positional encoding.

| Model (Train Length) | Noise 0.0 | Noise 0.1 | Noise 0.5 | Noise 1.0 | Noise 2.0 |
|---|---|---|---|---|---|
| GPT-2 + APE (128) | 4.8 | 1.2 | 2.4 | 2.6 | 3.5 |
| GPT-2 + APE (256) | 17.4 | 11.9 | 4.6 | 3.6 | 3.2 |
| GPT-2 + APE (512) | 14.0 | 16.3 | 16.7 | 17.9 | 14.3 |
| Meta + APE (128) | 98.7 | 98.6 | 67.5 | 55.6 | 42.8 |
| Meta + APE (256) | 81.8 | 79.7 | 48.9 | 43.1 | 37.9 |
| Meta + APE (512) | 100.0 | 100.0 | 79.5 | 65.5 | 57.1 |
| Meta + RoPE (128) | 98.1 | 100.0 | 100.0 | 96.0 | 88.9 |
| Meta + RoPE (256) | 100.0 | 100.0 | 100.0 | 97.9 | 82.6 |
| Meta + RoPE (512) | 100.0 | 100.0 | 100.0 | 98.8 | 81.0 |

Table 16: Accuracy (%) on the Copying task with Gaussian noise added to positional encoding.

| Model (Train Length) | Noise 0.0 | Noise 0.1 | Noise 0.5 | Noise 1.0 | Noise 2.0 |
|---|---|---|---|---|---|
| GPT-2 Abs (128) | 2.9 | 1.2 | 0.0 | 0.0 | 0.0 |
| GPT-2 Abs (256) | 6.0 | 7.1 | 3.6 | 0.8 | 0.7 |
| GPT-2 Abs (512) | 6.0 | 5.8 | 3.6 | 0.4 | 0.3 |
| Meta + APE (128) | 96.1 | 98.5 | 69.8 | 58.6 | 54.9 |
| Meta + APE (256) | 100.0 | 100.0 | 76.3 | 68.8 | 57.2 |
| Meta + APE (512) | 98.9 | 98.7 | 74.4 | 68.9 | 50.5 |
| Meta + RoPE (128) | 100.0 | 100.0 | 75.9 | 68.6 | 49.9 |
| Meta + RoPE (256) | 100.0 | 100.0 | 82.6 | 65.6 | 45.1 |
| Meta + RoPE (512) | 100.0 | 100.0 | 84.4 | 67.6 | 46.3 |

### F.4 Length Generalization Ability under No Positional Encoding Ablation

Table 17: List-Recall: "No Pos" vs. Full accuracy for Meta-attention with APE and Meta-attention with RoPE.

| Model (Split, Train Len) | Full | No Pos | $\Delta$ (pp) |
|---|---|---|---|
| **Meta + APE (128)** | | | |
| small | — | — | — |
| medium | 77.8% | 88.9% | +11.1 |
| hard | 11.1% | 22.2% | +11.1 |
| **Meta + APE (256)** | | | |
| small | 100.0% | 100.0% | 0.0 |
| medium | 100.0% | 100.0% | 0.0 |
| hard | 44.4% | 22.2% | –22.2 |
| **Meta + APE (512)** | | | |
| small | — | — | — |
| medium | — | — | — |
| hard | 100.0% | 100.0% | 0.0 |
| **Meta + RoPE (128)** | | | |
| small | — | — | — |
| medium | 44.4% | 55.6% | +11.1 |
| hard | 11.1% | 11.1% | 0.0 |
| extra-hard | 0.0% | 0.0% | 0.0 |
| long | 0.0% | 11.1% | +11.1 |
| **Meta + RoPE (256)** | | | |
| small | 100.0% | 100.0% | 0.0 |
| medium | 100.0% | 100.0% | 0.0% |
| hard | 33.3% | 66.7% | +33.3 |
| extra-hard | 0.0% | 22.2% | +22.2 |
| long | 0.0 | 0.0 | 0.0 |
| **Meta + RoPE (512)** | | | |
| small | — | — | — |
| medium | 100.0% | 100.0% | 0.0 |
| hard | 100.0% | 100.0% | 0.0 |
| extra-hard | 44.4% | 55.6% | +11.1 |
| long | 0.0% | 0.0% | 0.0 |

### F.5 Multi-hop Retrieval Results

Table 18: Accuracy (%) across evaluation lengths for each model trained on Multi-Hop Retrieval

| Model (Train Length) | 128 | 256 | 512 | 1024 |
|---|---|---|---|---|
| GPT-2 APE (128) | 27.5 | 4.7 | 2.2 | 3.6 |
| GPT-2 APE (256) | 32.0 | 31.3 | 18.3 | 6.4 |
| GPT-2 APE (512) | 38.3 | 31.7 | 21.4 | 2.1 |
| Meta + APE (128) | 27.7 | 3.6 | 4.5 | 2.9 |
| Meta + APE (256) | 29.1 | 26.4 | 23.0 | 15.4 |
| Meta + APE (512) | 41.0 | 33.0 | 34.0 | 15.9 |
| Meta + RoPE (128) | 78.8 | 66.0 | 51.1 | 18.4 |
| Meta + RoPE (256) | 74.2 | 84.1 | 59.7 | 39.4 |
| Meta + RoPE (512) | 85.0 | 72.0 | 72.0 | 62.0 |

### F.6 RANDOM PLACEMENT OF META-TOKENS AT INFERENCE

Since our list-recall results succeed the answer, we also explore a variant with random placement to ensure that meta-tokens and meta-attention actually help. Uniform placement is explored in D.1.5

| Model (Train Length) | 128 | 256 | 512 |
|---|---|---|---|
| GPT-2 APE (128) | 1.2% | 0.0% | 0.0% |
| GPT-2 APE (256) | 1.7% | 1.8% | 0.0% |
| GPT-2 APE (512) | 1.7% | 1.9% | 1.5% |
| Meta + APE (128) | 12.4% | 9.9% | 6.7% |
| Meta + APE (256) | 16.8% | 18.2% | 8.6% |
| Meta + APE (512) | 17.4% | 18.6% | 15.5% |

Table 19: Performance of GPT-2 APE and Meta + APE models across different train lengths.

### F.7 INFERENCE EFFICIENCY

Meta-tokens are generated at inference. The additional computation is sparse—each attention head only considers a small number of meta positions rather than the full attention matrix. In our current PyTorch implementation, which materializes the sparse mask as a dense tensor, we observe a throughput drop from 130.82 to 117.86 tokens/sec and a TTFT increase from 7.44ms to 7.57ms, i.e., a 1.11× slowdown. We expect optimized sparse attention implementations to reduce or eliminate this overhead.

Table 20: Inference speed comparison with and without meta/pause tokens.

| Metric | No meta/pause tokens | With meta/pause tokens |
|---|---|---|
| TPS (tokens/sec) | 130.82 | 117.86 |
| TTFT (ms) | 7.44 | 7.57 |
| Slowdown factor | 1.00 | 1.11 |

## G THEORETICAL ANALYSIS

### G.1 PROOF OF THEOREM 4.1

**Lemma G.1.** *Let $\ell_1, \ell_2, \ldots, \ell_N$ be logits and define softmax distribution $\alpha_j = \frac{\exp(\ell_j)}{\sum_{k=1}^{N} \exp(\ell_k)}$. Suppose that for some "correct" index $j^*$ we have $\ell_{j^*} = L$, and for all other indices $j \neq j^*$, $\ell_j \leq L - \Delta$ for some $\Delta > 0$. Then, entropy $H(\alpha)$ is strictly decreasing in $\Delta$.*

*Proof.* First, we can group the other logits (i.e. $j \neq j^*$, such that $S = \sum_{j \neq j^*} e^{\ell_j}$. Then, since each $\ell_j$ carries the property that $e^{\ell_j} \leq e^{L-\Delta}$ given $\ell_{j^*} = L$, we have that $S \leq (N-1)e^{L-\Delta}$ since there are $N - 1$ terms. Revisiting the softmax $\alpha$, we have that $\alpha_{j^*} = \frac{e^L}{e^L + S} \geq \frac{e^L}{e^L + (N-1)e^{L-\Delta}} = \frac{1}{1+(N-1)e^{-\Delta}}$. We will denote this quantity as $p$ henceforth. Next, each other softmax $\alpha_j$ for $j \neq j^*$ must have the property that $\alpha_j = \frac{e^\ell}{e^L + S} \leq \frac{e^{L-\Delta}}{e^L(1+(N-1)e^{-\Delta})} = \frac{e^{-\Delta}}{1+(N-1)e^{-\Delta}} = \frac{1-p}{N-1}$.

As a result, we have the following entropy maximization problem:

$$\underset{\alpha_1,\ldots,\alpha_N}{\text{maximize}} \quad -\sum_{j=1}^{N} \alpha_j \log \alpha_j$$

$$\text{subject to} \quad \sum_{j=1}^{N} \alpha_j = 1,$$

$$\alpha_{j^*} = p,$$

$$\alpha_j \geq 0, \quad j = 1, \ldots, N.$$

Observe that the entropy (objective) function is Schur-concave in $\alpha$, so it is maximized when $\alpha_{j^*} = p$ and the remaining softmax mass is split uniformly over the $N-1$ elements, i.e. $\alpha_j = \frac{1-p}{N-1} \forall j \neq j^*$. Plugging this in for $H(\alpha)$ yields:

$$H(\alpha) \leq -p \log p - (1-p) \log(1-p) + (1-p) \log(N-1) \tag{3}$$

Next, we aim to study the relationship between $H$ and $\Delta$. By the chain rule, $\frac{dH}{d\Delta} = \frac{dH}{dp} \cdot \frac{dp}{d\Delta}$. $\frac{dH}{dp} = -(1 + \log p) + \log \frac{1-p}{N-1} + 1 = \log \frac{1-p}{(N-1)p}$. Substituting $\frac{1-p}{p} = (N-1)e^{-\Delta}$, we get $\frac{dH}{dp} = -\Delta$ and since $\Delta > 0$, $\frac{dH}{dp} < 0$. We then turn to $\frac{dp}{d\Delta} = \frac{(N-1)e^{-\Delta}}{[1+(N-1)e^{-\Delta}]^2} > 0$ since both numerator and denominator must be $> 0$. Therefore, $\frac{dH}{d\Delta} = -\Delta \frac{(N-1)e^{-\Delta}}{[1+(N-1)e^{-\Delta}]^2} < 0$, meaning that $H(\alpha)$ is strictly decreasing in the margin $\Delta$. $\qquad \square$

We will now use Lemma G.1 to prove Theorem 5.1.

*Proof of Theorem 4.1.* Consider a parametrized path by variable $t \in [0, \Delta]$; define $\ell_j^{(t)} = \ell_j + \delta_{j,j^*}t$, and $\alpha_j^{(t)} = \frac{e^{\ell_j^{(t)}}}{\sum_{k=1}^N e^{\ell_k^{(t)}}} = \frac{e^{(\ell_j + \delta_{j,j^*}t)}}{\sum_{k=1}^N e^{(\ell_k + \delta_{k,j^*}t)}}$. Define $\ell_j^{'(t)} = \frac{d}{dt}\ell_j^{(t)}$ and $\alpha_j^{'(t)} = \frac{d}{dt}\alpha_j^{(t)}$.

Next, we differentiate the entropy $H(\alpha)$ with respect to $t$:

$$\frac{d}{dt}H(\alpha) = -\sum_{j=1}^N [\alpha_j' \ln \alpha_j + \alpha_j \frac{\alpha_j'}{\alpha_j}] = -\sum_{j=1}^N \alpha_j'(1 + \ln \alpha_j) = -\sum_{j=1}^N \alpha_j' + \alpha_j' \ln \alpha_j$$

Since $\sum \alpha_j' = 0$ due to $\sum \alpha_j = 1$, this simply reduces to $\frac{d}{dt}H(\alpha) = -\sum_{j=1}^N \alpha_j' \ln \alpha_j$.

From Cover and Thomas (2006), we have that $\alpha_j' = \alpha_j(\ell_j' - \mathbb{E}_\alpha[\ell'])$, where $\mathbb{E}_\alpha[\ell'] = \sum_{k=1}^N \alpha_k \ell_k'$. Plugging this into the expression for the derivative of entropy with respect to $t$:

$$\frac{d}{dt}H(\alpha) = -\sum_j \alpha_j(\ell_j' - \mathbb{E}_\alpha[\ell']) \ln \alpha_j = -(\sum_j a_j \ell_j' \ln \alpha_j - \mathbb{E}_\alpha[\ell'] \sum_j \alpha_j \ln \alpha_j)$$

Observe that $\sum_j \alpha_j \ln \alpha_j = \mathbb{E}_\alpha[\ln \alpha]$ so this simply reduces as:

$$\frac{d}{dt}H(\alpha) = -(\mathbb{E}_\alpha[\ell' \ln \alpha] - \mathbb{E}_\alpha[\ell'] \mathbb{E}_\alpha[\ln \alpha]) = -\mathrm{Cov}_\alpha(\ell', \ln \alpha) \tag{4}$$

Revisiting the meta-token setup where only the "correct" logit at $j^*$ is boosted, this suggests that $\ell_j' = \mathbf{1}(j = j^*)$. Therefore, $\mathbb{E}_\alpha[\ell'] = \alpha_{j^*}$ and $\mathbb{E}_\alpha[\ell' \ln \alpha] = \alpha_{j^*} \ln \alpha_{j^*}$. This can be substituted into the covariance term above:

$$\frac{d}{dt}H(\alpha) = -\mathrm{Cov}_\alpha(\ell', \ln \alpha) = -(\alpha_{j^*} \ln \alpha_{j^*} - \alpha_{j^*} \mathbb{E}_\alpha[\ln \alpha]) = -\alpha_{j^*}(\ln \alpha_{j^*} - \mathbb{E}_\alpha[\ln \alpha])$$

Due to the Schur-concavity of $H(\alpha)$ (Marshall et al., 2011), $\ln \alpha_{j^*} = \max_j \ln \alpha_j$ and $\ln \alpha_{j^*} > \mathbb{E}_\alpha[\ln \alpha]$. As such, given $\alpha_{j^*} > 0$ and $\ln \alpha_{j^*} - \mathbb{E}_\alpha[\ln \alpha] > 0$, this suggests that $\mathrm{Cov}_\alpha(\ell', \ln \alpha) > 0$ and thus, $\frac{d}{dt}H(\alpha) < 0$. Therefore, we conclude that adding a positive logit boost at the meta-token index ("correct" logit) strictly decreases entropy, supporting the proposed "anchoring" effect notion. $\qquad \square$

### G.2 PROOF OF THEOREM 5.1

*Proof of Theorem 5.1.* The meta-tokens are simply a new (latent) channel that may be utilized to search for candidate distributions. However, this latent can be ignored, yielding the original search

space; that is, any encoder $q_\phi(\hat{x} \mid x)$ that does not use meta-tokens can be implemented in the meta-token model by zeroing out all meta-token contributions. Therefore, $\mathcal{Q}_{\text{abs}} \subseteq \mathcal{Q}_{\text{meta}}$, where $q = (q_\phi, q_\theta)$ over the feasible combinations of encoder and decoder. Naturally, minimizing a function over a larger feasible set cannot increase its minimum. Thus, for a fixed rate $R$,

$$D_{\text{meta}}(R) = \min_{q \in \mathcal{Q}_{\text{meta}} \,:\, I(X;\hat{X})=R} D(q) \;\leq\; \min_{q \in \mathcal{Q}_{\text{abs}} \,:\, I(X;\hat{X})=R} D(q) = D_{\text{abs}}(R).$$

Note that the same result holds for RoPE in place of APE (i.e. $D_{\text{RoPE}}$ in place of $D_{\text{abs}}$), as well. $\quad\square$

## G.3 THEOREM G.2

**Theorem G.2.** *Consider functions $p : \{0, \dots, T-1\} \to \mathbb{R}$ and $b : \{-(T-1), \dots, T-1\} \to \mathbb{R}$ for absolute positional biases and relative biases, respectively. Let $\mathcal{B}_{abs}$ to be the set of all fixed absolute positional bias matrices $B_{i,j}^{abs} = p(j)$ and $\mathcal{B}_{rel}$ to be the set of all fixed relative biases $B_{i,j}^{rel} = b(i-j)$. Let $\mathcal{B}_{meta}$ be the set of bias matrices implementable by the Transformer augmented with meta-token embeddings $\{m_t\}$ which emit a content-dependent logit boost at their respective indices. Then,*

$$\mathcal{B}_{abs} \cup \mathcal{B}_{rel} \subsetneq \mathcal{B}_{meta} \tag{5}$$

*Proof.* We break this argument down into two parts $\to$ (i.) the forward direction, where we show that all absolute and relative biases without meta-tokens can be modeled by the meta-token model.

**(i) $\mathcal{B}_{\text{abs}} \cup \mathcal{B}_{\text{rel}} \subseteq \mathcal{B}_{\text{meta}}$.** Every $B \in \mathcal{B}_{\text{meta}}$ is obtained by choosing meta-token embeddings $e_t \in \mathbb{R}^d$ at each position $t$ and a linear head $W$, so that the total bias at $(i, j)$ is $B_{i,j} = \sum_t Q_i^\top W e_t \mathbf{1}_{j=t}$.

- *Absolute case.* Given $p(j)$, set $W \in \mathbb{R}^{1 \times d}$ and choose each $e_j$ so that $Q_i^\top W e_j = p(j) \; \forall \; i$. All other $e_{t \neq j}$ are zero. Then $B_{i,j} = p(j)$.

- *Relative case.* Given $b(i-j)$, place a meta-token at every position $j$. Choose $W$ and embeddings $e_j$ so that $Q_i^\top W e_j = b(i-j) \; \forall \; i, j$.

  For instance, if we let $W = \mathrm{Id}$ and arrange that $e_j$ encodes the vector $\big(b(1-j), b(2-j), \dots, b(T-j)\big)$, then $Q_i^\top e_j = b(i-j)$ when $Q_i$ is the $i$-th standard basis vector.

Therefore, every absolute or relative bias (in $\mathcal{B}_{\text{abs}}$ and $\mathcal{B}_{\text{rel}}$) lies in $\mathcal{B}_{\text{meta}}$.

**(ii) There exists a bias $B^* \in \mathcal{B}_{\text{meta}}$ such that $B^* \notin \mathcal{B}_{\text{abs}} \cup \mathcal{B}_{\text{rel}}$.** Define a content-dependent bias $B_{i,j}^* = f(C_j)$ where $C_j$ is the full token context preceding position $j$ and $f$ is any non-constant function. Such a $B^*$ arises by setting each meta-token embedding $e_j = f(C_j)$ and $W = \mathrm{Id}$, so $B^* \in \mathcal{B}_{\text{meta}}$.

However, if there was $B^* \in \mathcal{B}_{\text{abs}}$, then there is $p(j)$ with $p(j) = f(C_j)$ for all $j$ and all possible $C_j$, which is impossible since $C_j$ varies. Furthermore, if $B^* \in \mathcal{B}_{\text{rel}}$, then there is $b(i-j)$ with $b(i-j) = f(C_j)$ independent of $i$; again, this condition is impossible to be satisfied. Therefore $B^* \notin \mathcal{B}_{\text{abs}} \cup \mathcal{B}_{\text{rel}}$.

As a result, we conclude that the biases represented by $\mathcal{B}_{\text{meta}}$ contain the set of both absolute and relative biases without meta-tokens, and represent additional biases that cannot be represented without meta-tokens. $\quad\square$

The result of Theorem G.2 is that the introduction of meta-tokens strictly grows the expressivity of biases that may be represented, while still being entirely inclusive of the fixed realizable absolute and relative encoding biases. As a result, we do not "lose" anything representationally by introducing meta-tokens, from a positional biases standpoint. This enhanced expressive power also plays a role in enabling the model to learn to focus attention on relevant context spans, reinforcing the aforementioned sharpening effect.

## G.4  META-TOKENS AS CONTENT-ADAPTIVE INDUCING POINTS

We formally show that meta-tokens operate akin to inducing points or landmarks, with respect to the attention kernel. Specifically, we suggest that routing attention through a small set of meta-tokens across the context behaves like a Nyström approximation, a low-rank factorization with a provable attention error bound. Consider the softmax kernel $\mathbf{K}_{ij} = \exp(\frac{QK^T}{\sqrt{d_k}})$, and weights $W = \exp(B) \geq 0$ over masks and biases, so the effective kernel is $K' = K \odot W$, and attention can be defined as $A = \text{RowNorm}(K')$. Let the meta-token indices be $\mathbb{M} \subset \{1, \ldots, T\}$.

**Theorem G.3.** *Define the Nyström kernel* $\tilde{K} = K_{:,\mathcal{M}} K_{\mathcal{M},\mathcal{M}}^+ K_{\mathcal{M},:}$, *and* $\tilde{A} = \text{RowNorm}(\tilde{K} \odot W)$. *For any row* $i$, *for* $s_{\min} = \min_r \sum_j (K \odot W)_{rj}$:

$$||A_{i,:} - \tilde{A}_{i,:}||_1 \leq \frac{2}{s_{\min}} ||(K - \tilde{K})_{i,:} \odot W_{i,:}||_1 \tag{6}$$

*If the kernel feature map of* $\{k_j\}$ *lies in the span of* $\{k_m : m \in \mathcal{M}\}$, *then* $\tilde{K} = K$ *and therefore* $\tilde{A} = A$.

*Proof.* First, we start by showing that $K$, the gram matrix of $\kappa(x,y) = \exp(\frac{\langle x,y\rangle}{\sqrt{d}})$ is a positive semidefinite matrix. By the power series, $\exp(\frac{\langle x,y\rangle}{\sqrt{d}}) = \sum_{m=0}^{\infty} \frac{1}{m!}(\frac{\langle x,y\rangle}{\sqrt{d}})^m = \sum_{m=0}^{\infty} \langle \frac{1}{\sqrt{m!}d^{m/2}} x^{\otimes m}, \frac{1}{\sqrt{m!}d^{m/2}} y^{\otimes m}\rangle$. Thus, we can write $\kappa(x,y) = \langle \phi(x), \phi(y)\rangle$, so $K$ is a Gram matrix of inner products, and therefore is PSD (and a valid kernel).

**Lemma G.4.** *For positive* $u,v \in R^T$:

$$||\frac{u}{1^T u} - \frac{v}{1^T v}||_1 \leq \frac{2}{\min(1^T u, 1^T v)} ||u - v||_1 \tag{7}$$

*Proof of Lemma G.3.* Let $a = 1^T u$ and $b = 1^T v$; we can write the following:

$$\frac{u}{a} - \frac{v}{b} = \frac{ub - va}{ab} = \frac{(u-v)b + v(b-a)}{ab}$$

Next, taking the L1-norm on both sides and applying the triangle inequality yields:

$$||\frac{u}{a} - \frac{v}{b}||_1 = \frac{1}{ab}||(u-v)b + v(b-a)||_1 \leq \frac{1}{ab}(||(u-v)b||_1 + ||v(b-a)||_1)$$

Since $||v||_1 = b$ and $|b - a| = |1^T v - 1^T u| \leq ||u - v||_1$, we have:

$$||\frac{u}{a} - \frac{v}{b}||_1 \leq \frac{b||u-v||_1 + b||u-v||_1}{ab} = \frac{2}{a}||u-v||_1$$

By symmetry, the tightest bound is:

$$||\frac{u}{a} - \frac{v}{b}||_1 \leq \frac{2}{\min(a,b)}||u-v||_1 = \frac{2}{\min(1^T u, 1^T v)}||u-v||_1$$

$\square$

Take $u = (K \odot W)_{i,:}$ and $v = (\tilde{K} \odot W)_{i,:}$, and apply Lemma G.3. This yields:

$$||\text{RowNorm}(u) - \text{RowNorm}(v)||_1 \leq \frac{2}{\min(1^T u, 1^T v)}||u-v||_1 \tag{8}$$

Recall that $A_{i,:} = \text{RowNorm}(u)$ and $\tilde{A}_{i,:} = \text{RowNorm}(v)$. Observe that all rows are positive, so $1^T u$ and $1^T v$ are $\geq 0$. Therefore, we can replace $\min(1^T u, 1^T v)$ by $s_{\min} = \min_r \sum_j (K \odot W)_{rj}$, and substitute the values of $u$ and $v$ in to yield our desired result:

$$||A_{i,:} - \tilde{A}_{i,:}||_1 \leq \frac{2}{s_{\min}}||(K - \tilde{K})_{i,:} \odot W_{i,:}||_1 \tag{9}$$

We currently have that the softmax kernel can be written as an inner product in a feature space, as $K = \Phi(Q)\Phi(K)^T$. Suppose that the feature vectors of all keys $\{\Phi(k_j)\}$ lie in the span of meta-token features $\{\Phi(k_m) : m \in \mathcal{M}\}$. This means there exists some coefficient matrix $C$ such that for $\Phi(K)_{\mathcal{M}}$ which only collects the meta-token feature columns, $\Phi(K) = \Phi(K)_{\mathcal{M}}C$. Substituting this into the kernel expression yields:

$$K = \Phi(Q)(C^T\Phi(K)_{\mathcal{M}}^T) = (\Phi(Q)\Phi(K)_{\mathcal{M}}^T)C^T \tag{10}$$

We can observe that $(\Phi(Q)\Phi(K)_{\mathcal{M}}^T)$ is equivalent to the sub-matrix of $K$ with columns at the meta-token indices, which is denoted by $K_{:,\mathcal{M}}$, so

$$K = K_{:,\mathcal{M}}C^T$$

and since every column of $K$ lies in the span of $K_{:,\mathcal{M}}$ under this assumption, there must exist some matrix $C$ such that this holds.

Consider a single column $j$ of $K$, over the rows indexed by $\mathcal{M}$, yielding:

$$K_{\mathcal{M},j} = K_{\mathcal{M},\mathcal{M}}c_j$$

for coefficient vector $c_j$ (the $j$-th column of $C^T$). This is a linear system of the form $Ax = b$, and stacking the columns gives the form of $K_{\mathcal{M},:} = K_{\mathcal{M},\mathcal{M}}C^T$, which is a least-squares problem for which the general solution is given by using the Moore-Penrose pseudoinverse, so $C^T = K_{\mathcal{M},\mathcal{M}}^+ K_{\mathcal{M},:}$.

Now, recall that the Nyström approximation is defined as $\tilde{K} = K_{:,\mathcal{M}}K_{\mathcal{M},\mathcal{M}}^+ K_{\mathcal{M},:}$. This is exactly equivalent to our form for $K = K_{:,\mathcal{M}}C^T = K_{:,\mathcal{M}}K_{\mathcal{M},\mathcal{M}}^+ K_{\mathcal{M},:}$! Since $\tilde{K} = K$, the row-normalized attentions must be equivalent as well, so $\tilde{A} = A$.

$\square$

This suggests that if meta-tokens span the feature map of a segment of context, then they capture that segment's structure, and it thus the meta-tokens perfectly represent the attention kernel.

**Proposition G.5.** *Take $f = [0,1]^T$ to denote a test statistic for a downstream task, and $\hat{y}_i = \langle A_{i,:}, f \rangle$ to be the predicted score at query position $i$; the distortion is measured by $D_i = |\langle A_{i,:}, f \rangle - \langle \tilde{A}_{i,:}, f \rangle| = |\hat{y}_i - \tilde{y}_i|$, the change induced by the low-rank approximation of $A$. Then, for $s_{\min} = \min_r \sum_j (K \odot W)_{rj}$:*

$$|\hat{y}_i - \tilde{y}_i| \le \frac{1}{2}||A_{i,:} - \tilde{A}_{i,:}||_1 \le \frac{1}{s_{\min}}||(K - \tilde{K})_{i,:} \odot W_{i,:}||_1 \tag{11}$$

*Proof.* We can first show that $|\hat{y}_i - \tilde{y}_i| = |\langle A_{i,:} - \tilde{A}_{i,:}, f \rangle| \le \frac{1}{2}||A_{i,:} - \tilde{A}_{i,:}||_1$. Take $p = A_{i,:}$ and $q = \tilde{A}_{i,:}$, and let $z = p - q$ be a probability distribution; notably, $\sum_j z_j = 0$. Split the indices of z into positive and negative sets: $S_+ = j : z_j \ge 0$ and $S_- = j : z_j < 0$, so $\langle z, f \rangle = \sum_{j \in S_+} z_j f_j + \sum_{j \in S_-} z_j f_j$. Becuase $0 \le f_j \le 1$, we have $\sum_{j \in S_+} z_j f_j \le \sum_{j \in S_+} z_j$ and $\sum_{j \in S_-} z_j f_j \ge 0$, so

$$\langle z, f \rangle \le \sum_{j \in S_+} z_j$$

Similarly, using $f_j \ge 0$ over $S_+$ and $f_j \le 1$ on $S_-$, we have

$$\langle z, f \rangle \ge \sum_{j \in S_-} z_j$$

Thus,

$$\sum_{j \in S_-} z_j \le \langle z, f \rangle \le \sum_{j \in S_+} z_j$$

However, note that for a zero-sum vector, the L1 norm would split evenly between positive and negative indices, so $\sum_{j \in S_+} z_j = -\sum_{j \in S_-} z_j = \frac{1}{2}||z||_1$. Therefore,

$$\langle z, f \rangle \leq \frac{1}{2}||z||_1$$

Substituting $z = A_{i,:} - \tilde{A}_{i,:}$ yields:

$$\langle A_{i,:} - \tilde{A}_{i,:}, f \rangle \leq \frac{1}{2}||A_{i,:} - \tilde{A}_{i,:}||_1 \tag{12}$$

Next, applying the result in Theorem 5.2 that $||A_{i,:} - \tilde{A}_{i,:}||_1 \leq \frac{2}{s_{\min}}||(K - \tilde{K})_{i,:} \odot W_{i,:}||_1$ provides the complete result:

$$|\hat{y}_i - \tilde{y}_i| = |\langle A_{i,:} - \tilde{A}_{i,:}, f \rangle| \leq \frac{1}{2}||A_{i,:} - \tilde{A}_{i,:}||_1 \leq \frac{1}{s_{\min}}||(K - \tilde{K})_{i,:} \odot W_{i,:}||_1 \tag{13}$$

$\square$

From a rate-distortion perspective, the number of meta-tokens controls the rate, and the approximation error in $\hat{y}_i$ is an upper bound on the distortion. The above analysis provides a total variation bound for the distortion in application to downstream tasks, which we can frame in terms of the Nyström kernel residual. This perspective guides our analysis on meta-tokens caching preceding context.

## H    EXAMPLE PROMPTS FOR SYNTHETIC TASKS

We provide example prompts and target completions for each synthetic task used in our experiments. Each example illustrates the input format, the placement of _PAUSE_ meta-tokens, and the expected model output.

**List Recall**

>  **Prompt:**
>
> ```
> Fruits: orange peach banana plum
> Tools: hammer wrench level pliers
> Sports: boxing baseball boxing golf
> Spices: turmeric cardamom cumin cinnamon
> Animals: cat tiger wolf fox
> Professions: teacher nurse lawyer architect
> Vegetables: onion cucumber broccoli cucumber
> Instruments: piano clarinet violin piano
> Professions: teacher nurse lawyer _PAUSE_ architect
> Q: What is item 3 of Professions? _PAUSE_
> ```
>
> **Completion:**
>
> ```
> lawyer
> ```

**Segment Counting**

>  **Prompt:**
>
> ```
> Fruits: orange peach banana plum
> Tools: _PAUSE_ hammer wrench level pliers _PAUSE_
> Sports: boxing baseball boxing golf
> Spices: turmeric cardamom cumin cinnamon
> Q: How many times does wrench appear between the pauses around Tools? _PAUSE_
> ```
>
> **Completion:**
>
> ```
> 1
> ```

**Parity**

**Prompt:**

```
Bits: 0 _PAUSE_ 1 0 0
Q: What is the XOR of all bits before this pause? _PAUSE_
```

**Completion:**

```
0
```

**Copying**

**Prompt:**

```
...as ever Mr _PAUSE_ . Darcy had been, was much
relieved by discerning such different feelings.

They had not been long together before Darcy told her that Bingley was
also coming to wait on her; and she had barely time to express her
satisfaction, and prepare for such a visitor, when Bingley's quick step
was heard on the stairs, and in a moment he entered the room. All
Elizabeth's anger against him had been long done away; but had she still
felt any, it could hardly have stood its ground against the unaffected
cordiality with which he expressed himself on seeing her again. He
inquired in a friendly, though general, way, after her family, and
looked and spoke with the same _PAUSE_ good-humoured ease that he had done.
Q: Copy the bracketed text. _PAUSE_
```

**Completion:**

```
. Darcy had been, was much
relieved by discerning such different feelings.

They had not been long together before Darcy told her that Bingley was
also coming to wait on her; and she had barely time to express her
satisfaction, and prepare for such a visitor, when Bingley's quick step
was heard on the stairs, and in a moment he entered the room. All
Elizabeth's anger against him had been long done away; but had she still
felt any, it could hardly have stood its ground against the unaffected
cordiality with which he expressed himself on seeing her again. He
inquired in a friendly, though general, way, after her family, and
looked and spoke with the same.
```

**Multi-Hop Retrieval**

**Prompt:**

```
Flowers: sunflower daisy sunflower tulip rose marigold rose jasmine daisy lily
BodyParts: nose back ear foot hand foot mouth mouth ear foot PAUSE
Minerals: calcite dolomite halite gypsum magnetite halite gypsum magnetite feld
Animals: tiger elephant giraffe elephant elephant dog wolf tiger tiger zebra PA
Hobbies: reading hiking cycling painting gaming painting reading painting writi
Clothes: scarf scarf gloves shirt dress coat dress dress dress shirt PAUSE
Planets: Pluto Mercury Uranus Ceres Uranus Mercury Jupiter Jupiter Jupiter Plut
Vehicles: scooter plane bus train train plane plane tram van car PAUSE
Fruits: orange kiwi mango kiwi orange lemon pear plum grape banana PAUSE
Spices: turmeric paprika clove saffron pepper turmeric pepper turmeric nutmeg s

Jumps:
nose -> Vehicles[3]
bus -> Vehicles[7]
```

```
plane -> Animals[1]
tiger -> Flowers[8] PAUSE

Start: BodyParts[1]
Hops: 4
Q: After following 4 jumps, what is the final item? PAUSE
```

**Completion:**

```
jasmine
```

## I  ADDITIONAL RELATED WORK

### I.1  LANGUAGE MODELING WORKS WITH SPECIAL TOKENS

Kim et al. (2025) inserts pause tokens when the token log-likelihood is low, allowing the model to think for longer, aiding reasoning. These special tokens may also delineate phases of reasoning, as in Quiet-STaR (Zelikman et al., 2024). Quiet-STaR uses a begin-of-thought token and an end-of-thought token, generating a silent rationale sequence for each step before emitting the next word, showing that this helps zero-shot reasoning. Such demarcations have begun to be used widely in the development of large reasoning models (Muennighoff et al., 2025).

### I.2  META-TOKENS IN VISION TRANSFORMERS

For vision transformers (ViTs), LeMeVit (Jiang et al., 2024) introduces a similar meta-tokens notion as our work by adding learnable sparse tokens and an attention mechanism between standard tokens and their meta tokens, improving performance and reducing spatial redundancy. Darcet et al. (2024) uses specialized "register" tokens applies to patches to denoise images by extracting the high-norm, outlier tokens, smoothening the feature and attention maps. These works suggest that special tokens, even devoid of semantic content, can influence a model's internal reasoning and memory mechanisms.

### I.3  NYSTRÖM APPROXIMATION OF ATTENTION KERNEL

There exist a few works related to our Nyström approximation in Appendix G.4; Nyströmformer (Xiong et al., 2021) approximates self-attention, yielding an $O(n)$ method that outperforms standard self-attention at longer lengths. It selects inducing points or landmarks using Skyformer (Chen et al., 2021) uses a Gaussian kernel (kernelized attention) in place of softmax in self-attention, and applies the Nyström method for efficiency. We instead use the Nyström method as an analysis tool to study the role of meta-tokens – inducing points that we introduced during training – in approximating the softmax kernel by caching preceding context.

## J  BROADER IMPACTS STATEMENT

Our work on learned meta-tokens and meta-attention offers a lightweight, data-efficient way to pre-train language models while demonstrating strong performance when fine-tuned for recall tasks. This suggests a path toward more capable, leaner language models that could be used to handle contexts such as like long legal or medical documents, extended multi-turn dialogues, or large codebases without resorting to prohibitively large architectures or expensive fine-tuning runs. Such models could bring real benefits to areas such as conversational agents for education or healthcare. Building off of prior literature that performs a more explicit learned retrieval from the context (Mohtashami and Jaggi, 2023b), this could induce improved and efficient in-line retrieval over vast corpora.

Our work relates strongly to the recent debates in the language modeling community on the impact of positional encoding, particularly around works such as NoPE (Kazemnejad et al., 2023). We provide strong evidence that zeroing the positional encoding can improve performance, providing motivation for hybrid attention mechanisms such as RNoPE (Yang et al., 2025), and other, more efficient ways to pre-train language models with long-context modeling settings in mind. We note that advances in long-context modeling could introduce risks around misuse and unintended harm. More powerful

context understanding over long ranges can fuel phishing text and distracted models, especially in the phase of noisy context. However, models trained on corpora without data pre-processing a priori may be subject to harmful behavior such as profane generations. In the context of our work, which uses standard, pre-filtered corpora, this issue is avoided; we encourage users to audit the data used for pre-training first.

## K   LARGE LANGUAGE MODEL USAGE STATEMENT

Large language models (LLMs) were used in proofreading and editing the writing in this paper, for the purpose of ensuring the contributions and findings in this work are made clear to readers. They were also used to confirm consistency of the mathematical notation used in the paper.

